# The Role of Endothelial Progenitor Cells in Atherosclerosis and Impact of Anti-Lipemic Treatments on Endothelial Repair

**DOI:** 10.3390/ijms23052663

**Published:** 2022-02-28

**Authors:** Velimir Altabas, Lora Stanka Kirigin Biloš

**Affiliations:** 1Department of Endocrinology, Diabetes and Metabolic Diseases, Sestre Milosrdnice University Hospital Center, 10000 Zagreb, Croatia; 2School of Medicine, University of Zagreb, 10000 Zagreb, Croatia; lora.s.kirigin@gmail.com

**Keywords:** endothelial progenitor cells, statins, fibrates, PCSK9 inhibitors, angiopoietein-like protein inhibitors, LDL apheresis

## Abstract

Cardiovascular complications are associated with advanced atherosclerosis. Although atherosclerosis is still regarded as an incurable disease, at least in its more advanced stages, the discovery of endothelial progenitor cells (EPCs), with their ability to replace old and injured cells and differentiate into healthy and functional mature endothelial cells, has shifted our view of atherosclerosis as an incurable disease, and merged traditional theories of atherosclerosis pathogenesis with evolving concepts of vascular biology. EPC alterations are involved in the pathogenesis of vascular abnormalities in atherosclerosis, but many questions remain unanswered. Many currently available drugs that impact cardiovascular morbidity and mortality have shown a positive effect on EPC biology. This review examines the role of endothelial progenitor cells in atherosclerosis development, and the impact standard antilipemic drugs, including statins, fibrates, and ezetimibe, as well as more novel treatments such as proprotein convertase subtilisin/kexin type 9 (PCSK9) modulating agents and angiopoietin-like proteins (Angtpl3) inhibitors have on EPC biology.

## 1. Introduction

Atherosclerosis is a vascular disease caused by the build-up of plaques in the innermost layers of arteries, leading to arterial wall thickening and hardening, and narrowing of the arterial lumina [1]. Non-modifiable risk factors of atherosclerosis include advanced age and male sex, while modifiable risk factors, including arterial hypertension, diabetes, obesity, physical inactivity, smoking, and hypercholesterolemia, are globally addressed in cardiovascular prevention programs. Other important risk factors, including chronic kidney disease, familial hypercholesterolemia, various endocrine disorders or a previous history of cardiovascular events, may require a more specific intervention [2,3].

The initial event in the pathogenesis of atherosclerosis is endothelial injury and dysfunction, followed by mononuclear adhesion and migration into the arterial subendothelial space [1,4]. Furthermore, oxidative stress and insulin resistance stimulate the overproduction of proinflammatory cytokines and other inflammatory mediators resulting in a state of virtually permanent low-grade inflammation. Smooth muscle cells migrate to the arterial intima, differentiate into fibroblasts, and produce matrix molecules like elastin and collagen, eventually leading to plaque growth and formation of the fibrous caps. Fibrous caps may rupture, exposing the underlying extremely thrombogenic core. Such events prompt further thrombus formation and release of more inflammatory mediators, resulting in arterial stenosis or occlusion [1]. This leads to end organ damage, and depending on the anatomic site of the injured vessel, presents as myocardial infarction, ischemic stroke, or critical limb ischemia [5,6,7].

In general, atherosclerosis is still viewed as an irreversible process, strongly linked to aging. Advanced atherosclerotic lesions both in humans and animals consist of necrosis, calcification, and fibrosis, making lesion regression and possible dissolution unlikely [8], despite various conservative and invasive treatment options.

The discovery of endothelial progenitor cells (EPCs) at the end of the twentieth century provided new insights to the pathophysiology of atherosclerosis and offered new prospects in treatment. Due to the unique characteristics of these cell lines, the idea that atherosclerosis regression was possible emerged. These myeloid derived cells are capable of virtually endless division and differentiation into healthy and functional endothelial cells at the site of vessel injury, repairing the vessel wall, and promoting neovascularization [9,10]. This review focuses on the role of endothelial progenitor cells in atherosclerosis, and the impact standard as well as novel lipid-lowering treatments have on EPC-related vascular repair.

## 2. Endothelium Biology and Function

The endothelium was once seen as a simple barrier between the blood and the vascular wall to prevent blood extravasation. It is now perceived as a metabolically active tissue important for healthy vessel function. The human endothelium exceeds the size of many organs, with a surface area of more than 800 m^2^, and weight of approximately 1500 g. More than 250 biologically active substances are produced and secreted by endothelial cells, some responsible for vascular tone regulation, cell adhesion, thromboresistance, smooth muscle cell proliferation, and inflammation. In addition, anticoagulant, antiplatelet and fibrinolytic factors are synthesized in these cells. Thus, the endothelium is a large endocrine and paracrine organ interplaying with virtually all other tissues and organs [11,12].

Vasoconstriction and vasorelaxation are important features of the endothelium, responsible for optimal blood supply to end organs and tissues, maintaining cell respiration and other metabolic demands. The endothelium itself produces vasoactive molecules affecting vascular tone locally, interplaying with circulating vasoactive mediators [13,14].

One of the most important vasoactive substances released by the endothelium is nitric oxide (NO). NO was first identified and named after its vasorelaxant properties as endothelium-derived relaxing factor. It is synthesized in endothelial cells from L-arginine in the presence of endothelial NO synthase and several cofactors. Once released, NO acts locally by penetrating smooth muscle cells in the vascular wall leading to guanylate cyclase activation and cyclic guanosine monophopsphate (cGMP)-mediated vasodilatation [15]. NO also inhibits the synthesis of vascular cell adhesion molecule-1 (VCAM-1) and monocyte chemoattractant protein-1 (MCP-1), resulting in decreased expression of nuclear factor κB (NFκB), which action is further impeded by oxidative phosphorylation in mitochondria and S-nitrosylation of cysteine residues of NFκB. NF-κB target genes in endothelial cells are vascular cell adhesion molecule-1 (VCAM-1), intercellular adhesion molecule-1 and E–selectin. These substances are responsible for extravasation and are linked to inflammation, thus aggravating endothelial dysfunction [16].

Guanylate cyclase activation leading to vasodilation can also be achieved by other distinct signaling molecules like bradykinin, adenosine, vascular endothelial growth factor (during hypoxia), serotonin (during platelet aggregation), prostaglandins, C-type natriuretic peptide, and gaseous molecules like carbon monoxide (CO), and hydrogen sulfide (H_2_S) [16,17]. Other important molecules involved in vasomotion synthesized in endothelial cells are endothelin, vasoconstricting prostanoids, prostacyclin, and angiotensin convertase at the endothelial cell surface. Prostacyclin, in addition, blocks platelet thrombus formation and hinders clotting [16,18]. Vasomotion is also mediated by intercellular potassium ion accumulation due to endothelium–derived hyperpolarizing factor, changing vascular electric conductivity and enhancing propagation of electrical signals along blood vessels affecting gap junctions and smooth muscle cells in the arterial wall [14].

The anticoagulant properties of the endothelium are linked to the synthesis of thrombomodulin and heparin sulfate proteoglycan, and the secretion of tissue factor pathway inhibitor. All these substances interfere with the coagulation cascade, inhibiting thrombus formation and/or enhancing fibrinolysis [19].

## 3. Pathophysiology of Atherosclerosis

The hallmark of atherosclerosis is the progressive build-up of atherosclerotic plaques. Plaques develop slowly through a complex series of cellular events within the arterial wall, influenced by a variety of local vascular circulating factors. Plaques typically develop in the arterial intima, although atherosclerotic changes may involve all three layers of the arterial wall [1,20].

There are two basic theories on atherosclerosis development, the response-to-injury theory and the response to retention theory. Both theories see endothelial injury as an early event preceding other developments [1]. In the response-to-injury hypothesis, the initial injury due to mechanical or chemical factors leads to endothelial dysfunction and allows inflammatory cells to penetrate the arterial wall, followed by fat accumulation and proliferation of smooth muscle cells [21]. Not only are major arteries involved, but vasa vasorum, blood vessels supplying oxygen and nutrients to the artery wall of mother arteries may also be affected [21]. In the response-to-retention theory, lipoprotein retention in response to predisposing mechanical strain and cytokines in the vessels’ extracellular matrix is the triggering event further resulting in atherogenesis [22]. Following endothelial injury, fat is accumulated in deeper regions of the subendothelial layer, forming lipid pools, rich in proteoglycans and hyaluronan, covered with a layer of vascular smooth muscle cells [1,23].

Nonetheless, plaques develop in regions with nonlaminar blood flow, as in artery arches, branches, or curvatures. Blood flow alterations are detected by endothelial cells’ blood flow sensing organelles, primary cilia, leading to a myriad of functional and structural alterations. In these microlocations endothelial cell lining exhibits different, cuboidal morphology compared to regions with laminar blood flow, where the cells are commonly aligned in the direction of blood flow [24]. On the molecular level, this phenotype of endothelial cells shows epigenetic changes due to altered DNA methylation, with activated pro-inflammatory NF-κB pathways, as well as suppressed protective factors like Kruppel-like factor 4 (KLF4), and impaired production of NO [24,25]. At the same time, the endothelial barrier is more permeable to lipoproteins, making it vulnerable to low density lipoprotein (LDL) accumulation and cell migration [1]. Endothelial cells in nonaffected regions express a more anti-inflammatory and anti-thrombotic phenotype and are organized differently [26].

Inflammatory cells and fatty deposits interplay in a vicious way, with fat causing further mononuclear cell migration and activation into monocyte-derived macrophages in the vessels’ intima, and pro-inflammatory cells aggravate endothelial dysfunction allowing easier fat accumulation [1]. The most important contributing factor is oxidized lipoprotein particles in the vessel wall, originating from LDL particles in blood plasma. A complex set of biochemical reactions regulates the oxidation of LDL, involving enzymes such as lipoprotein associated phospholipase A2 (Lp-LpA2), and free radicals produced through oxidative stress in the endothelium. The entire process is accelerated by low levels of high-density lipoprotein (HDL), which removes excess cholesterol from peripheral tissues and carries it back to the liver. Activated macrophages secrete pro-inflammatory mediators like tumor necrosis factor alpha (TNFα), interleukin-1 (IL-1), and interleukin-6 (IL-6) [20,27]. At this stage, endothelial cells secrete less NO and more vasoconstrictive cytokines like endothelin 1 and molecules like VCAM, ICAM, and monocyte chemotactic protein 1 (MCP-1). These vasoconstrictive and adhesive molecules further enhance the adherence and migration of monocytes to the injured vessel wall [1,20]. Monocyte derived macrophages then ingest cholesterol, resulting in foam cell formation. As the process progresses, these cells create fatty streaks, visible on the artery wall in early atherosclerosis. This process is still reversible, as fatty streaks can disappear under certain conditions. Foam cells may become apoptotic, enhancing inflammation locally [28].

Another event is the migration of smooth muscle cells from the artery’s muscle layer into the vascular intima due to cytokines secreted by the damaged endothelial lining and present foam cells. Smooth muscle cells further proliferate and ingest lipids in an already pro-inflammatory area, where locally acting growth factors, oxidized low density lipoprotein, and homocysteine contribute to plaque build-up [1,29]. Furthermore, smooth muscle cells can transform into chondrocytes, osteocytes, adipocytes, or macrophage-foam cells, depending on the local environment [30]. The bulk of these lesions is made of excess fat, collagen, and elastin. The fibrous cap contains smooth muscle cells, providing stability to the whole structure. As more fat accumulates, the plaque size increases, progressively changing the vessels’ architecture. No apparent narrowing is present at this stage, but blood flow may become more turbulent, increasing shear stress on the vessels’ wall, causing further endothelial microlesions, and perpetuating the entire process of atherogenesis [20,24].

Later, the proliferation rates of smooth muscle cells slow down, possibly due to increased expression of cell cycle inhibitors like p16 and p21, and impaired response to growth factors. In addition, their phenotype changes from contractile to a more synthetic. Dysregulated production of pro-inflammatory cytokines, growth factors, and extracellular matrix modifiers occurs, accelerating the process of vascular remodeling [31]. These mature smooth muscle cells release pro-inflammatory cytokines and matrix metalloproteinases affecting the collagen fraction of the plaque, making the plaque more vulnerable to rupture. Pro-inflammatory cytokines are able to change the phenotype of activated macrophages, promoting M1 and M4 subtypes, linked to plaque instability [32,33]. These advanced atherosclerotic plaques may undergo necrotic changes in their core and fibrinoid tissue. As cells die, calcifications may occur. The exact mechanisms of plaque calcifications are still unclear [34].

Stenosis due to plaque enlargement is a late event, which may even never occur, or it may be clinically asymptomatic if the blood flow is not significantly compromised. Dramatic and life-threatening complications like infarction are linked to plaque erosion or rupture, sometimes without previous symptomatic stenosis. This triggers injured endothelial cells to excrete excess thrombotic factors (e.g., von Willebrand factor [VWF] and thromboxane A2 [TXA2]), and decreased amounts of antithrombotic factors (e.g., heparin) leading to clot formation and enlargement [35]. Complete vessel obstruction within a short time may occur resulting in tissue ischemia and necrosis [36,37]. A schematic summary of atherosclerosis pathophysiology and its clinical correlates is shown in Figure 1.

## 4. Endothelial Repair

Exposure to various cardiovascular risk factors may result in functional and structural endothelial damage, ranging from delicate metabolic alterations in endothelial cells to cell loss by apoptosis. Damaged vessels lose their vasoactive ability due to impaired synthesis of vasoactive substances, but also because of increased rigidity due to structural changes of the vessel wall. Endothelial dysfunction may be the first step leading to more serious conditions like accelerated atherosclerosis and associated vascular complications [16,38].

Basically, vessel integrity can be restored if the inherent reparatory mechanisms are functional [9,10,39]. These mechanisms involve cell replication and replacement of unfunctional endothelial cells [40]. First, already existing mature endothelial cells may undergo mitotic processes, but because they are mostly terminally differentiated cells, their ability to proliferate is rather low [41,42]. The second mechanism leading to repair of the damaged endothelial lining is mediated through circulating EPCs. These immature cells have the capacity to proliferate and differentiate into mature endothelial cells. They originate from the bone marrow, and some other tissues like the spleen, liver, or fat. They circulate in the blood stream and may adhere to the damaged endothelium. Once they are embedded in the damaged endothelium, they proliferate and differentiate into mature functional and structural endothelial cells [9,10,41].

Circulating EPCs are in various stages of differentiation. There are at least two distinct types of cells: the early (less differentiated) and late (better differentiated) EPC [9]. Distinct features of early EPC include a spindle shape cell phenotype, their colony forming units (CFU), and the presence of several harboring markers (CD31, CD34, CD45, CD133, Tie2). Late EPCs in addition express vascular endothelial growth factor receptor-2 (VEGF-R2), vascular endothelial (VE)- cadherin and vWF and have a cobblestone shape. Late EPCs can produce nitric oxide. The maturation level of these cells influences their role in vascular repair. The role of less differentiated cells is mostly restricted to their paracrine function, providing growth factors, while more differentiated cells are able to provide more mature cells needed for actual vascular cellular repair [9,10,43].

The entire process of vascular repair mediated through EPCs consists of several distinct events like progenitor cell mobilization from their organ of origin, circulation in the blood stream, harbouring at the place of damaged endothelium (“homing”) and finally, further differentiation and cell maturation [10]. Different substances are involved in the mobilization of progenitor cells from their place of origin into the blood stream, like NO, but also growth factors and cytokines, including vascular endothelial growth factor (VEGF), stromal-cell-derived factor-1α (SDF-1α), impaired glucose metabolism, erythropoietin, thyroid hormones and estrogens [44,45,46]. Once in the blood stream, EPCs migrate towards damaged endothelial regions where they adhere to the damaged vessel surface. This process is significantly enhanced by certain molecules like stromal derived factor (SDF)-1α. The concentration of SDF-1α is upregulated in the damaged endothelium due to tissue hypoxia. SDF-1α interacts with CX chemokine receptor 4 (CXCR4) on the endothelial surface [46]. After being embedded in the injured endothelium, progenitor cells proliferate and maturate and thus physically replace damaged mature endothelial cells. In addition, they synthesize and excrete vasculogenic cytokines and growth factors enhancing replication of already present mature endothelial cells [47].

Mounting evidence indicates that major cardiovascular risk factors interfere with different aspects of endothelial progenitor cell biology (mobilization, homing, differentiation and function) [16,47,48,49,50,51]. Consequently, EPCs’ ability for vascular repair seems to be decreased in patients with advanced atherosclerosis [52].

## 5. Endothelial Repair in Patients with Lipid Disorders

Decreased EPC numbers and their impaired replicatory and migratory properties are seen in several cardiovascular risk factors including diabetes mellitus, arterial hypertension, lipid disorders, smoking, physical inactivity, and unhealthy eating habits [18,47]. Hypercholesterolemia is linked to mechanical endothelial injury and dysfunction. In the context of endothelial repair, there is accumulating evidence that hypercholesterolemia may reduce the availability and function of EPCs, thus limiting vascular repair [50].

Delivery of cholesterol-rich lipoproteins to the endothelium is an important process in the pathogenesis of atherosclerosis. It is influenced by lipoprotein type and concentration, and the integrity of the endothelium. Importantly, LDL cholesterol may induce vascular endothelial cell apoptosis, due to its increased toxicity after being oxidized to oxidized LDL (ox-LDL) within macrophages and change the permeability of the endothelial barrier by inducing inflammation. A vicious cycle with the interplay of LDL arterial wall retention, inflammation, smooth muscle cell proliferation, macrophage activation, and coagulation irregularities is involved [1,20,28]. Ox-LDL has been shown to impair proliferation, migration, and adhesion capacity of EPCs. This has been explained by the activation of transcriptional regulator NF-κB [53]. Interestingly, sexual dimorphism of ox-LDL concentrations has been reported [54], affecting EPC differently [55], at least in mice.

In contrast to LDL cholesterol’s deleterious effects in the pathogenesis of atherosclerosis, high density lipoprotein (HDL) cholesterol has been shown to be protective [56]. There is far less data about its impact on EPC health. So far, HDL has been shown to improve the viability of early EPC, and to a lesser extent their functionality, in terms of adhesion properties [57]. Dysfunctional HDL does not benefit EPC biology [58,59].

Triglycerides have raised less interest in this field in comparison to hypercholesterolemia. However, elevated triglycerides contribute to overall cardiovascular risk. Hypertriglyceridemia has also been shown to negatively affect EPC biology. Hypertriglyceridemia leads to endothelial dysfunction and injury by interfering with SDF-1/CXCR-4 binding and NO pathways, thus affecting mobilization, migration, homing, and the vasculogenic properties of EPC [60,61].

Changes in EPC biology in dyslipemic states may help us better understand how well-established and newer therapeutic strategies can prevent, delay and possibly reverse atherosclerosis. For that reason, some contemporary trials correlated treatment effects on EPC biology using endothelial function tests. To date, there is a myriad of invasive and non-invasive methods to assess endothelial function in addition to cell cultures, and specific endothelial biomarkers. Some of them are used to evaluate vascular tone modulation and tissue perfusion, others to assess dynamic permeability, or anticoagulation and fibrinolysis [62]. Flow-mediated dilation (FMD) is an imaging technique used for endothelial vasomotion assessment. Other methods used to assess vascular dilation include laser-based techniques, venous occlusion plethysmography and finger plethysmography [62,63,64,65]. In addition to endothelium-dependent vasodilation, arterial stiffness (compliance) determination and pulse wave analysis can provide more insight into vessel health, but it is not recommended for routine clinical use [66]. Furthermore, the anticoagulant and fibrinolytic properties of the endothelium can be determined. Tissue plasminogen activator inhibitor, factor X and thrombin blood levels in basal circumstances, after stimulation with various substances (e.g., substance P for tissue plasminogen activator inhibitor) or in cell cultures can give valuable information on endothelial status [62,67].

Biomarkers of endothelial function include different molecules like angiopoietins (angiopoietin 1 and 2), selectins like intercellular adhesion molecule-1 (ICAM-1), vascular cell adhesion molecule-1 (VCAM-1), and platelet endothelial cell adhesion molecule-1 (PECAM-1), growth factors like VEGF and its soluble VEGF receptor-1 (VEGFR-1), and platelet derived growth factor (PDGF), together affecting angioneogenesis, inflammation and endothelial permeability. Additionally, endothelial breakdown products such as syndecan-1, chondroitin sulfate, dermatan sulfate, serum hyaluronic acid, and heparan sulfate are markers of endothelial injury and dysfunction [62]. Similarly, increased counts of circulating endothelial cells (CECs) originating from the mature endothelium were observed in people with cardiovascular risk factors and acute myocardial infarction. Thus, CEC can be considered as a marker of endothelial injury and dysfunction [68,69]. Furthermore, small membranous particles released from endothelial cells named endothelial microvesicles correlate with endothelial dysfunction in different states like infections, cancer, and autoimmune diseases [70]

## 6. Standard Treatment for Lipid Disorders: Statins, Ezetimibe and Fibrates

Statins have been used for decades in the treatment of hypercholesterolemia and became the fundamental therapy of atherosclerosis and its complications. Statins improve outcomes in primary and secondary cardiovascular prevention and have a central place in modern guidelines for atherosclerosis and cardiovascular disease treatment [71,72,73]. Their primary mode of action is inhibition of 3-hydroxy-3-methylglutaryl-coenzyme A (HMG-CoA) reductase, the key enzyme responsible for cholesterol synthesis, but their beneficial effects reach beyond cholesterol lowering. Even before any notable changes in lipid concentrations are observed, a number of pleiotropyic effects including anti-inflammatory, anti-oxidative, anti-thrombotic and profibrinolytic, increased endothelial NO production, and antiapoptotic actions, alleviate atherosclerotic progression [74,75]. In addition, favorable effects on EPC biology are reported. Statin therapy has been associated with increased circulating EPCs due to enhanced mobilization, differentiation, and increased longevity, as well as enhanced homing to sites of vascular injury and re-endothelization via enhanced expression of on EPC cell surface adhesion molecules [75].

Substance-specific effects on EPC biology are shown in Table 1. Clinical correlates are shown where appropriate.

Statins were shown to increase the endothelial progenitor cell blood count as early as 1 week after treatment initiation, reaching a plateau within 3 to 4 weeks. This effect and further differentiation are mediated through NO synthesis pathways, which increases CXCR4 expression on the surface of circulating EPCs [98]. In addition, statins decrease micro non-coding RNA levels named miR 221 and 222 leading to up-regulated EPC differentiation and mobilization [99]. Statins also interact with the phosphoinositide 3-kinase (PI3K)/protein kinase B (Akt)/mammalian target of rapamycin (mTOR) pathway resulting in increased VEGF levels affecting angiogenesis directly and indirectly through increased NO levels [74,75]. Finally, statins modulate oxidative stress by diminishing oxLDL production in vessel walls and thus enhancing EPC numbers, mobilization, function and ability to migrate and or integrate into vasculature [75].

Studies have shown favorable results for different statins, like simvastatin, pravastatin, pitavastatin, atorvastatin and rosuvastatin, indicating the effect on EPCs could be a class effect [75,91,93,95,97]. Interestingly, even herbal remedies containing lovastatin, later produced as a first generation statin, positively impact EPC biology [40]. Increased EPC count induced by statin therapy remains stable [100]. The favorable statin-mediated effects on EPC count, may be dose-dependent, at least for atorvastatin. Higher EPC counts were observed in patients receiving 80 mg of atorvastatin as compared to lower doses in various populations. In addition, atorvastatin reloading in patients receiving moderate dose statin therapy and undergoing percutaneous coronary intervention, triggered an acute increase in EPC count, and benefited their functionality, while decreasing inflammatory markers like high sensitivity C reactive protein (hCRP) [81,101]. Furthermore, a decreased 30-day adverse event rate in NSTEMI and unstable angina patients was observed [102]. In some studies there was a favourable effect on FMD and endothelial biomarkers [79,80,81,82,83,84,90,92,96,97]. It must be noted that statin side effects may hamper their use and limit potential therapeutic benefits in patients with atherosclerosis. The most important side effects include myopathy, liver damage, drug-induced diabetes, and neurological disturbances, and may lead to therapy discontinuation. Caution should be used in elderly people or in patients with chronic kidney disease since pharmacodynamic and pharmacokinetic drug properties may differ from those of the general population [103].

Ezetimibe is usually prescribed when cholesterol levels are not well controlled with statin monotherapy, or as monotherapy in specific patients when statins are contraindicated or not well tolerated. Ezetimibe impairs cholesterol absorption in the gut, targeting the Niemann-Pick C1-Like1 (NPC1L1) sterol transporter, which is responsible for the intestinal absorption of cholesterol and phytosterols [104]. Data suggests that ezetimibe does not benefit EPC biology. Ezetimibe even showed negative effects on the endothelium, increasing circulating endothelial microparticles [105], indicating enhanced apoptosis of endothelial cells. In addition, when used in combination with simvastatin, no further improvements in EPC count was found in patients with coronary heart disease, suggesting that treatment benefits are not related to EPC biology [93].

Fibrates are another class of antilipemic drugs that are widely used. They stimulate peroxisome proliferator activated receptor (PPAR) alpha, affecting gene expression involved in triglyceride and cholesterol metabolism. They have been shown to reduce triglyceride levels, and to a lesser extent, LDL levels, while increasing HDL concentration. A single published study showed beneficial effects of fenofibrate on EPCs in cell cultures obtained from patients with chronic heart failure [106].

In summary, evidence suggests that statins have a pronounced beneficial effect on EPC count and function, which is probably independent of their lipid-lowering effect, ezetimibe is ineffective, and data on fibrates are still limited.

## 7. Novel Treatments for Lipid Disorders: Proprotein Convertase Subtilisin/Kexin Type 9 (PCSK9) Modulating Agents and Angiopoietin-like Proteins (Angtpl3) Inhibitors

PCSK9 modulating agents form a new class of anti-lipemic drugs, suitable as add-on treatment to statins and ezetimibe, but also as monotherapy in statin and/or ezetimibe intolerant patients at increased cardiovascular risk [107]. By structure, they are either PCSK9 monoclonal antibodies interfering with circulatory PCSK9 (alirocumab, evolocumab) or small interfering ribonucleic acids (siRNA), modulating PCSK9 synthesis (inclisiran) [108,109].

PCSK9 acts via a canonical pathway to reduce LDL-receptor (LDL-R) recycling in the liver, thus decreasing LDL-R bioavailability. PCSK9 binds to LDL-Rs on the cell surface, resulting in receptor degradation, lowering the number of disposable LDL-Rs. Consequently, circulating LDL cannot be properly removed from the blood, and LDL concentrations rise. PCSK9 is also expressed in other tissues and organs, like the intestine, kidneys, and blood vessels. PCSK9 of kidney and blood vessel origin is secreted into the blood and downregulates LDL-R levels at other cells, including hepatocytes and macrophages, decreasing LDL clearance. Furthermore, it seems that PCSK9 enhances the migratory capacity of monocytes and inhibits reverse cholesterol transport in macrophages, favoring foam cell formation in atherosclerotic plaques. PCSK9 expressed in smooth muscle cell vessel walls was shown to promote inflammation and contribute to endothelial cell apoptosis through the Bcl-2/Bax–Caspase9–Caspase3 mitochondrial pathway and the p38/Jun N-terminal kinases/mitogen-activated protein kinases (p38/JNK/MAPK) signaling pathway, disrupting endothelial integrity, and resulting in endothelial dysfunction and atherosclerosis development [108].

PCSK9 inhibition is highly effective in reducing LDL cholesterol levels, with a decrease of 60% from baseline seen within days. PCSK9 inhibitors reduce plaque size, measured by intravascular ultrasound and serial magnetic resonance [110,111]. Furthermore, a significant reduction in cardiovascular risk was demonstrated for alirocumab and evolocumab in large international blinded randomized trials [112,113].

Considering EPC biology, a recent cross-sectional clinical study in humans indicated beneficial effects. Namely, endogenous PCSK9 levels were inversely correlated with circulating EPC count in patients with type 2 diabetes mellitus on statin therapy, as well as in the entire cohort of patients. No correlation was found in patients not taking statins [114]. Furthermore, a small clinical study demonstrated favorable effects of PCSK9 inhibitors, alirocumab and evolocumab, on EPC biology in patients with coronary artery disease. There was a significantly higher EPC count and proliferative capacity in patients treated with PCSK9 inhibitors, detected as early as one month after therapy initiation. Increased VEGF levels accompanied the effect of PCSK9 inhibitors. The study was too small to examine the role of evolocumab and alirocumab separately [115]. Possible side effects of alirocumab and evolocumab include flu like symptoms and local injection site reactions, giving a more favourable safety profile in comparison to statin therapy [116].

The impact of inclisiran, a small interfering mRNA molecule inhibiting the translation of PCSK9, on EPC biology has not been investigated so far.

Angiopoietins are growth factors with a prominent role in embryonal and adult vasculogenesis. There are some molecules closely related to angiopoietin with a different impact on blood vessels [117,118,119]. Angiopoietin-like protein 3 (Angtpl3), a protein secreted by the liver, raised interest as a potential target for lipid lowering drugs, because loss of function mutations was shown to be protective in terms of lipid derangements, atherosclerosis, and cardiovascular risk. There are several gene variants; carriers of loss of function variant develop the phenotype of familial hypolipidemia and those with complete Angtpl3 deficiency have lower triglycerides and LDL cholesterol, and raised HDL cholesterol, and are prone to longevity [120,121]. Considering lipid metabolism, Angptl3 enhances the cleavage of lipoprotein lipase (LPL) by proprotein convertases in target tissues, leading to LPL dissociation from the cell surface. In addition, Angptl3 inhibits in vitro endothelial lipase (EL) activity, increasing HDL catabolism. The mechanism of action involves a complex of Angptl3 with the related protein Angptl8, that promotes Angptl3 effects [120,122,123]. Beyond lipid-lowering, inhibition of Angptl3 was shown to improve endothelial function [124]. This effect may be mediated indirectly through improved lipid levels, but also directly by binding endothelial integrin v3, and by stimulating Wnt/-catenin signaling [12].

Evinacumab, a recombinant monoclonal Angptl3 antibody, was recently approved for the treatment of familial hypercholesterolemia. It is highly effective, leading to a reduction of LDL cholesterol by 50% in patients with refractory familial hypercholesterolemia, independent of the LDL receptor, with overall mild side effects, like flu like symptoms and injection site reactions [125]. Other Angptl3 inhibitors, including antisense oligonucleotide (ASO) are at different stages of clinical trials [12].

A single study of Angptl3 inhibition on neoangiogenesis demonstrated that Angptl3 enhances cell to cell adhesion via integrin αvβ3 and migration of endothelial cells [126]. This research was done in the early era of stem cell research, on human microvascular venous endothelial cells, and not EPCS, but activation of the same pathways was later confirmed to be important for improving EPC biology [127].

Recently, a hypothesis pointing to beneficial effects of Angptl3 on EPC biology was published, but has not been confirmed [128]. In short, evidence regarding the impact of Angptl3 inhibition on EPCs is lacking, and future high-quality research is needed.

## 8. Plasma Apheresis

LDL apheresis acutely removes circulatory LDL particles by extracorporeal filtration. It was first introduced in the 1970s, and has been used for patients with familial hypercholesterolemia unresponsive to statin therapy, with and without documented atherosclerotic vascular disease [129,130]. There is still no consensus regarding the frequency of this procedure, with reported intervals ranging from once weekly to once in two months [131].

LDL apheresis improves circulating PCSK9 and Lp(a) levels, and upregulates LDL-Rs in tissues, enhancing statin sensitivity [132]. Patients with familial hypercholesterolemia treated by LDL apheresis and statins for over one year, show coronary plaque area reduction and an increase in arterial luminal diameter [133].

A recent study demonstrated additional beneficial effects of LDL apharesis on blood lipids and EPC count in patients with percutaneous coronary interventions for acute coronary syndrome. All patients were assigned to moderate-to high-intensity statin therapy with either atorvastatin ≥40 mg or rosuvastatin ≥20 mg, regardless of being in the study or control group [134]. There was an additional robust acute decline in LDL levels at hospital discharge in patients treated with apheresis, but there was no significant difference between groups noted after 30 days. Interestingly, a more sustained mobilization of endothelial progenitor cells was noted up to three months after randomization, peaking 30 days after the coronary procedure. There were marginal changes in coronary artery architecture in terms of reduced nonculprit coronary plaque size in patients treated with LDL apheresis. This trial had a relatively small sample size, short follow-up, and was not powered to assess clinical outcomes. However, changes in non-culprit coronary plaque size was comparable to similar trials in patients with acute coronary syndrome [134].

In short, LDL apheresis results in fast and profound LDL reduction and increases EPC count. LDL apheresis may mobilize EPC in a durable fashion, but this method is accompanied by adverse effects related to the method itself like access-related complications (bleeding, fistula infection), device-related complications, hypotension and rarely, arrhythmia [135].

## 9. Conclusions

Endothelial repair mediated by endothelial progenitor cells (EPCs) has raised substantial interest in the scientific community. Small clinical trials have shown the beneficial effect several standard and novel lipid-lowering treatments have on EPC biology. Statins are widely studied, and clinical data suggest that their pleiomorphic, rather than their lipid-lowering properties impact EPC count and function. At least for some statins this effect is dose-dependent. However, side effects and statin intolerance, and in some cases their insufficient efficacy may limit their benefits in real life thus requiring add-on therapies. So far, ezetimibe has not shown any beneficial impact on EPC biology, data for fibrates are still scarce.

Considering newer anti-lipemic drugs and treatments like PCSK9 inhibitors, Angplt3 antibodies and LDL apheresis, initial results are promising, but further research is needed to determine their mode of action and effectiveness on EPC count, and migratory and proliferative potential.

## Figures and Tables

**Figure 1 ijms-23-02663-f001:**
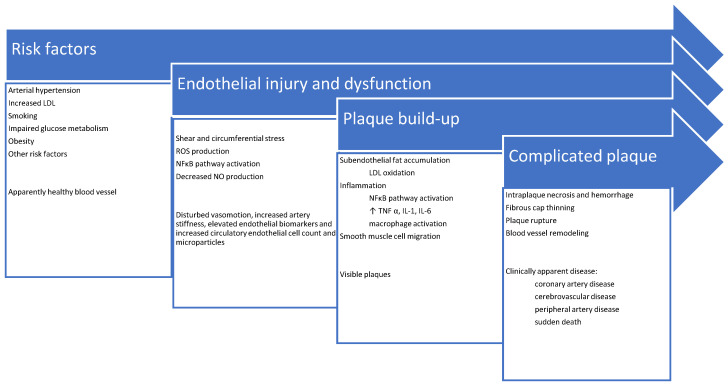
Schematic summary of atherosclerosis pathophysiology and its clinical presentations.

**Table 1 ijms-23-02663-t001:** Summary of clinical trials on various antilipemic drugs affecting EPC biology.

Substance	Study Population	Major Findings in Study Groups
Atorvastatin reloading with 80 mg [76,77]	53 patients on long-term statin treatment who underwent percutaneous coronary interventions (PCI)	↑ EPC count ↑ EPC-CFU
Atorvastatin 80 mg vs. atorvastatin 10 mg preloading [78]	20 statin-naïve male patients undergoing angiography	3.5-fold increase in EPC levels in the 80 mg group
Atorvastatin 40 mg vs. atorvastatin 10 mg [79]	26 patients with ischemic heart failure	↑ EPC ↑ FMD ↓ TNF-α
Atorvastatin 40 mg vs. atorvastatin 10 mg vs. placebo [80]	58 patients with coronary heart disease	↑ EPC in both atorvastatin groups ↓ VEGF and CRP
Atorvastatin 80 mg reloading vs. 40 mg vs. no statin [81]	45 patients undergoing coronary angioplasty	↑ early EPCs in 80 mg group ↓ increase in cardiac troponin
Atorvastatin 40 mg vs. 10 mg [82]	100 patients with ischemic cardiomyopathy	↑ EPC ↓ hsCRP, oxLDL
Atorvastatin 40 mg vs. control group [83]	108 patients with coronary slow flow	↑ EPCs ↑ EPC adhesion, migration and proliferation ↑ NO ↓ hs-CRP, ET-1 and IL-6
Atorvastatin 40 mg vs. placebo [84]	60 consecutive patients who underwent isolated, first-time CABG	↑ early EPCs ↓ hsCRP Less atrial fibrillation
Atorvastatin 20 mg vs. placebo [85]	50 patients undergoing elective coronary surgery	↑ EPCs
Atorvastatin 80 mg vs. atorvastatin 20 mg [86]	40 ST-segment elevation myocardial infarction (STEMI) patients undergoing PCI	↑ EPCs
Atorvastatin 20 mg vs. placebo [87]	68 patients with chronic pulmonary heart disease	↑ EPCs
Atorvastatin 20 mg vs. no statin [88]	48 patients with a first-time non-lacunar ischaemic stroke	↑ EPC increment EPC increment ≥4 CFU-EC predicted favorable clinical outcome
Rosuvastatin 40 mg [89]	26 patients with mixed dyslipidaemia	↑ EPC count ↑ EPC-CFU
Rosuvastatin 10 mg vs. placebo [90]	60 patients with systolic heart failure	↑ EPC FMD, VEGF, fibrinogen, MMP-9, IL-6, IL-1β, oxLDL, PerOx, NT-proBNP, and uric acid levels did not correlate with EPC level
Rosuvastatin 40 mg vs. placebo [91]	42 patients with chronic heart failure (CHF)	↑ EPC ↑ FMD
Rosuvastatin 10 mg vs. no treatment [92]	32 hypercholesterolemic patients	↑ EPC ↑ FMD
Simvastatin 80 mg vs. simvastatin 20/10 mg ezetimibe [93]	68 patients with coronary artery disease	no effect on EPC
Simvastatin 80 mg mono-treatment with combination treatment of 10 mg simvastatin and 10 mg ezetimibe [94]	19 obese men with the metabolic syndrome	↑ EPCs regardless of study group
Pravastatin 40 mg vs. placebo [95]	20 healthy postmenopausal women	↑ EPC-CFU
Pravastatin 10 mg vs. placebo [96]	29 patients with isolated low HDL cholesterol	↑ EPC ↑ FMD
Pitavastatin 2 mg vs. atorvastatin 10 mg [97]	26 patients at high cardiovascular risk	↑ EPC ↑ eNOS expression ↑ adhesion ability of early EPCs ↑ migration and tube formation capacities of late EPCs

↑ for increased, ↓ for decrease

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
