# Peer review of "The Role of Endothelial Progenitor Cells in Atherosclerosis and Impact of Anti-Lipemic Treatments on Endothelial Repair"

_ijms, 2022, doi:10.3390/ijms23052663_

Round 1

Reviewer 1 Report

The authors discuss in an adequate sequence. Nevertheless, the conclusion was weak. They did not include statin, such as their essential use in public health, which is vital given their low cost and allied with its effectiveness. In comparison, antibodies hold significant promise for those with resistance to lower levels of atheromatous versus the high prices. However, both therapies impact the biology of endothelial progenitors associated with hyperlipidemia.
The authors are missing a discussion item on the hormonal influence on endothelial progenitor cells, such as control thyroid function in combating metabolic syndrome in the biology of these cells.

Author Response

Dear reviewer, 

we appreciated your comments on the manuscript: "The role of endothelial progenitor cells in atherosclerosis and impact of anti-lipemic treatment on endothelial progenitor cells"

As suggested, we have sharpened the statements in the conclusion regarding statin treatment, and have added thyroid hormones disturbances as a cause of aberrations in EPC biology with relevant references. 

However, since the second reviewer asked for some more extensive changes of the paper's structure, the entire manuscript will look much different compared to the previous version. We tried to do our best, and we hope you will be satisfied with the changes. 

Please find a rewritten version of the manuscript in the attachment. 

Reviewer 2 Report

The text of the first four subchapters is divided into many paragraphs, I recommend revising it to a more comprehensive form for better reading and understanding. Please include the figure or scheme of summarization of the described pathophysiological mechanisms leading to atherosclerotic changes. Given the current trend of non-invasive assessment of endothelial function/dysfunction in clinical practice, I recommend including a separate section regarding the assessment of endothelial function.

The second part of the manuscript concerns lipid disorders and their standard and novel treatment. In this context, I recommend revising the title of subchapter 7 as “Novel treatment…” (similarly to Chapter 6 "Standard treatment…."). Further, I recommend including a summary table of the findings of the recent studies. It would be useful to include a separate section on contraindications or side effects.

I recommend checking for typos and to unify References format

Author Response

Dear reviewer, 

thank you for your suggestions in order to improve the manuscript: "The role of endothelial progenitor cells in atherosclerosis and impact of anti-lipemic treatments on EPC repair". 

We have substantially reduced to number of paragraphs in the first four chapters.

A scheme explaining atherosclerosis progression was added to the paper.

A separate section regarding assessment of endothelial function was added in chapter 5. 

We added a chapter named "Novel treatments for lipid disorders: proprotein convertase subtilisin/kexin type 9 (PCSK9) modulating agents and angiopoietin-like proteins (Angtpl3) inhibitors" instead of former separate chapters discussing PCSK9 inhibitors and angiopoietin inhibitors.

A table summarizing findings in relevant clinical studies on different drugs affecting EPC biology was added. Since there were no separate studies for alirocumab, avolocumab and evinacumab and EPC, these substances were only discussed in the text. 

Side effects are discussed for drugs with any kind of proven impact on EPC biology.

We hope that you will find the paper much better now and suitable for publication. 

Best regards.  

Round 2

Reviewer 2 Report

I recommend checking (correcting) the format of the References.